# Microbiota Dynamics of Mechanically Separated Organic Fraction of Municipal Solid Waste during Composting

**DOI:** 10.3390/microorganisms9091877

**Published:** 2021-09-03

**Authors:** Vladimir Mironov, Anna Vanteeva, Diyana Sokolova, Alexander Merkel, Yury Nikolaev

**Affiliations:** Research Center of Biotechnology, Winogradsky Institute of Microbiology, Russian Academy of Sciences, 119071 Moscow, Russia; very-well1966@mail.ru (A.V.); sokolovadiyana@gmail.com (D.S.); alexandrmerkel@gmail.com (A.M.); nikolaevya@mail.ru (Y.N.)

**Keywords:** composting, mechanically separated organic fraction of municipal solid waste, microbial community, biodiversity, functional genes

## Abstract

Mechanical-biological treatment of municipal solid waste (MSW) facilitates reducing the landfill workload. The current research aimed to study general activity parameters, content, functions, and diversity of fungal and prokaryotic microbiota in mechanically separated organic fraction of MSW (ms-OFMSW) composting, without using bulking agents and process-promoting additives. During 35 days of composting, vigorous emission of CO_2_ (max. 129.4 mg CO_2_ kg^−1^ h^−1^), NH_3_ (max. 0.245 mg NH_3_ kg^−1^ h^−1^), and heat release (max. 4.28 kJ kg^−1^ h^−1^) occurred, indicating intense microbial activity. Immediately following the preparation of the composting mixture, eight genera of lactic acid bacteria and fungal genera *Rhizopus*, *Aspergillus*, *Penicillium*, *Agaricus*, and *Candida* were predominant. When the temperature increased to more than 60 °C, the microbial biodiversity decreased. Due to succession, the main decomposers of ms-OFMSW changed. The Bacillaceae family, the genera *Planifilum*, *Thermobifida*, and *Streptomyces*, and the fungal genera *Thermomyces* and *Microascus* were involved in the processes of organic matter mineralization at the high-temperature and later stages. The biodiversity of the microbiota increased at the stages of cooling and maturation under conditions of relatively high nitrogen content. Thus, the microbial community and its succession during ms-OFMSW composting were characterized for the first time in this work.

## 1. Introduction

It is estimated that by 2050 the production global of municipal solid waste (MSW) will grow to 3.4 billion tons per year [1]. The disposal of MSW in landfills is an unavoidable reality and remains the predominant waste processing method regardless of a country’s income.

MSW is subjected to mechanical-biological treatment (MBT) [2], including (i) recovery of recyclable fractions and separation of materials with a high calorific value for the production of fuel from waste [3]; (ii) mechanical separation of a fraction ≤70 mm containing >50% of the biodegradable organic fraction (OFMSW); and (iii) aerobic high-temperature microbiological stabilization (composting) of mechanically separated OFMSW (ms-OFMSW) [4,5]. As a result, biostabilized residual waste (BSRW) is obtained [6], which is further subjected to disposal at landfills [7]. MBT of waste reduces the emission of harmful and fetid substances into the atmosphere, the formation of dangerous filtrates from the landfill body, and the volume of waste disposed of, thereby increasing the duration of landfill use [6].

OFMSW comprises 42 to 75% of the total MSW content [1]. The composition of OFMSW depends on various factors, such as the season, climate, and geographical location, but, on average, OFMSW consists of about 30–69% carbohydrates (starch, cellulose, and hemicellulose), 5–10% proteins, and 10–40% lipids [5].

Successful composting of organic substrates occurs if suitable conditions are provided for the microbial community development: optimal humidity, oxygen access, temperature, and stirring [8]. At the initial stage, the organic matter (OM) decomposers are mainly bacteria and fungi. During the subsequent thermophilic phase, the population of fungi reduces. At the final stage of composting, fungi and bacteria that decompose recalcitrant OM appear. Microbial composition and diversity affect the level of mineralization and microbiota activity, especially in stressful conditions (high temperature, presence of toxic compounds) [9,10]. The more diverse the microbial community, the more resistant it becomes. High microbial diversity implies the presence of a significant number of different types of destructors that are mutually beneficial and contribute to the simultaneous decomposition of various organic substances by the community. High biodiversity in the high-temperature composting stage promotes the effective biodegradation of organic matter, and deep stabilization of the compost is promoted during cooling and maturation [10].

Metabolic changes in microorganisms involved in the bioprocess lead to changes in respiration intensity [11] and heat release. Respirometry is often used to assess aerobic processes in solid-phase cultivation and indicates the degree of potential mineralization or decomposition of the material [11,12]. In the absence of high microbial activity, the respiration rate is low; therefore, the material is more stable. The calculation of the respirometric index is recommended by the European Commission to confirm the stability of waste [12].

It should be noted that OFMSW composting has been studied in sufficient detail on model substrates with various bulking agents or a mixture of food and plant waste (grass, leaves) [11,13,14]. In our opinion, that does not fully correspond to actual waste, such as ms-OFMSW. Few studies have investigated composting of ms-OFMSW obtained at MBT plants. Thus, this research is urgently needed.

Numerous studies are devoted to the respirometrical estimation of the general biological activity during OFMSW composting [6,11,12], and the bacterial community and its relationship with the physicochemical parameters of composting [6,7,14]. However, there is insufficient research showing the development of both prokaryotic and fungal components of microbiota combined with environmental factors during ms-OFMSW composting. Understanding the succession of the total microbiota (bacteria, archaea, fungi) will deepen knowledge about ms-OFMSW composting and improve the waste processing technology, such as via the creation of an effective inoculum. Therefore, the data on the microbial community of ms-OFMSW will form the basis for future studies on the isolation of enrichment and pure microbial cultures, and the creation of active consortia and their usage as starting cultures to enhance degradation in general or of particular compounds (e.g., plastic).

The current research aimed to study the general biological activity indicators, composition, function, and biodiversity of fungal and prokaryotic communities to determine the effectiveness of ms-OFMSW composting without the introduction of bulking agents, or chemical or microbial additives.

Therefore, the objectives of this work were as follows: (i) to investigate the composition of microbial communities predominant at each stage of ms-OFMSW composting and the presence of functional genes in the course of composting; (ii) to assess the relationship of the microbial diversity and composition with environmental parameters and indicators of biological activity (biogenic element flows and heat release).

## 2. Materials and Methods

### 2.1. Composting Material

Ms-OFMSW from an MBT plant (Dolgoprudny city district, Moscow region) served as a substrate for composting. At the MBT plant, incoming MSW was sorted on drum separators with 70 mm cells into two fractions: (1) a wet fraction with a size less than or equal to 70 mm with a high content of biodegradable waste (ms-OFMSW); (2) a dry fraction >70 mm, enriched with materials having a significant calorific value, mainly plastic. Samples of ms-OFMSW were obtained at the industrial composting plant of the Grunt Eco company in the Moscow region (coordinates of the point: 55.540794, 38.079914). Waste samples were collected for two days in June 2020, having a total volume of about 100 L. The composition of the sample is shown in Appendix A.

The two fractions were crushed separately using an industrial shredder SHR 220–600 (Tula Machines, Tula, Russia) with 20 mm grid cells. We used the homogenization recommendations of Lacorte et al. [15] during laboratory experiment preparation and sampling stages. After mixing the crushed fractions, 30 dm^3^ of the material was placed in three containers of 10 dm^3^ each for composting.

### 2.2. Experimental Setup

The composting process was carried out for 98 days in laboratory conditions using an experimental setup comprising a multi-chamber bioreactor for aerobic solid-phase biodegradation of organic waste, according to the scheme described earlier [16].

The experimental setup was operated as follows. A container with ms-OFMSW having a volume of 10 dm^3^ (4836 ± 657 g) was placed in a chamber. The containers were made of a polymer material with 5 mm perforations in the walls and bottom to provide aeration. The temperature in each chamber was maintained independently at the current substrate temperature of ±0.2 °C using a heating element and an IVTM-7/2S temperature regulator (Eksis, Moscow, Russia).

The substrate was aerated for 98 days with air of ambient temperature and consumption of 0.04 L min^−1^ kg^−1^ dry OM using a compressor and RMS-A-0.035 GUZ-2 rotameters (Pribor-M, Korolyov, Russia). The gas composition of the substrate atmosphere was determined daily with an MAG-6 S-1 gas analyzer (Eksis, Moscow, Russia): CO_2_, NH_3_, and H_2_S. The temperature mode in the substrate was monitored continuously. Tap water was added to the substrate to compensate for losses caused by evaporation so that the water content was maintained at 60%. Moistening was carried out on the sampling days after taking substrate samples. There were three replicates of the experiment.

### 2.3. Determination of the Composted Mass Parameters

Samples for physicochemical and molecular microbiological studies were taken on the day of the experiment start (day 0) and then on days 7, 14, 21, 28, 56, and 98. For molecular microbiological analysis, 200 mL of the substrate was taken from each container from different places; the resulting combined sample of 600 mL was homogenized and sieved through a KP-131 sieve (Ruspribor, Saint Petersburg, Russia) having 0.5 mm cells. The latter fraction (about 100 mL) was ground with a pestle in a mortar. Then, a sample of 1.5 mL was placed in a sterile Eppendorf-type tube and frozen at a temperature of −20 °C.

The values of pH, electrical conductivity (EC), water content, organic matter (OM), total organic carbon (C), NH_4_^+^ and NO_3_^−^ content, germination index (GI), total Kjeldahl nitrogen (N), and C/N were measured and calculated as described previously [17]. The water-soluble compounds, pH, EC, and GI were determined from a suspension of the sample (10 g) in 300 mL distilled water [18]. Measurements of pH and EC were taken using an ANION 4150 laboratory analyzer (Ifraspak-Analit, Novosibirsk, Russia). The water content (%) was determined by the thermogravimetric method on an EVLAS-2M humidity analyzer (Sibagropribor, Novosibirsk, Russia). The OM content was determined using the thermogravimetric method at 430 °C [19] using a PM-16M-1200 muffle furnace (EVS, Saint Petersburg, Russia). The total organic carbon (%) was estimated as OM:1.8 [20]. The soluble NH_4_^+^ and NO_3_^−^ were determined on a Hach Lange DR 5000 spectrophotometer (Hach, Düsseldorf, Germany) according to the manufacturer’s manual: NH_4_^+^ was measured with Nessler’s reagent, and NO_3_^−^ was measured by the method based on cadmium reduction. The Kjeldahl total nitrogen (N, %) in the dry matter was determined by sample mineralization in a DKL 6 automatic digester (Velp Scientifica, Usmate, Italy) under conditions of heating up to 420 °C with concentrated sulfuric acid in the presence of hydrogen peroxide, and a mixed catalyst with subsequent distillation of ammonium into a boric acid solution using an UDK 139 semiautomatic distillation system (Velp Scientifica, Usmate, Italy) and titration with hydrochloric acid on an Easy Plus titrator, model Easy pH (METTLER TOLEDO, Greifensee, Switzerland). The C/N ratio was calculated using experimental data. The effect of BSRW on plant growth was assessed by seed germination and the root length of *Raphanus sativus* based on the germination index (GI, %) calculation [21]. The measurements were performed in three replicates. Initial substrate parameters are given in Table 1.

### 2.4. Heat Release

The thermal energy spent on the substrate self-heating was calculated according to a well-known expression, also applicable in the practice of composting calculations [24,25], kJ kg^−1^ h^−1^:q = (Cc (T2 − T1))/τ,(1)
where: T1, T2—initial and final substrate temperature, respectively, °C; τ—time between measurements, h.

The calculated weighted average value of the ms-OFMSW heat capacity was found considering the heat capacity of the components and their content in the mixture (Appendix A) and was 1.68 kJ kg^−1^ °C^−1^. The published data for OFMSW confirm the obtained value of the heat capacity: 2.18–2.50 kJ kg^−1^ °C^−1^ [26]; 1.86–4.09 kJ kg^−1^ °C^−1^ [27]; 1.67–3.90 kJ kg^−1^ °C^−1^ at a bulk density of 350–725 kg m^−3^ [28].

### 2.5. Profiling of Prokaryotic and Fungal Communities Based on 16S rRNA GENE and Internal Transcribed Spacer (ITS)

The composition of the microbial community was analyzed by the number of 16S rRNA genes and ITS copies. The processes of DNA isolation, PCR, and Illumina MiSeq high-throughput sequencing of the 16S rRNA gene region and ITS were performed by general techniques and were described in our earlier work [17]. For each sample, isolation of DNA was undertaken without replication; for each DNA sample, library preparation and sequencing were undertaken in duplicate.

The primary processing of the raw reads was carried out as described earlier [29]. All 16S rRNA gene sequence reads were then processed by the SILVAngs 1.3 pipeline [30] using the default settings: 98% similarity threshold was used for creating operational taxonomic units (OTUs) tables; 93% was the minimal similarity to the closest relative that was used for classification (other reads were assigned as “No Relative”). All ITS sequence reads were processed by the Knomics-Biota [31] using the “ITS fungi” pipeline, where the reads were classified by mapping against the UNITE database version 7.2 (QIIME release, version, accessed date: 1 December 2017) using the BWA-MEM algorithm (BWA version 0.7.12-r1039). A 97% similarity threshold was used for creating OTUs. The 16S rRNA gene library of each sample consisted of 5000 reads, and ITS library consisted of 10,000 reads.

Search for the genes of different degradation pathways was carried out using the KEGG [32] online service. Based on the data on the representation of taxa in the libraries, putative physiological and biochemical properties of the prokaryotic communities were determined using the iVikodak software package [33]. The contribution of various groups of prokaryotes to the metabolic pathways during the period of the highest potential functional activity (day 21 of composting) was also evaluated using the KEGG database. The heatmaps of the most abundant (>1% in respect to prokaryotic/fungal biota) community members at the genus level and heatmap representing physiological and biochemical properties were constructed using the online resource ClustVis [34]. Clusterization of samples shown in the genera distribution heatmap was performed in ClustVis using correlation distance and average linkage. Chao1 [35] and Shannon indices were calculated for prokaryotic and fungal communities by the SILVAngs 1.3 pipeline and Knomics-Biota “ITS fungi’’ pipeline, respectively.

### 2.6. Statistical Data Analysis

Throughout the study, three replicates of each measurement were performed. The data were subjected to analysis of variance (ANOVA) using the least significant difference test. The test of significance was determined at *p* < 0.05. The results are presented in the form of an “average value ± standard deviation” of three replicates. The Pearson correlation coefficient (r) was used to determine the relationships between different variables. The significance of the Pearson correlation coefficients was tested at α = 0.05 [36].

## 3. Results

### 3.1. Primary Composting Parameters

#### 3.1.1. Temperature Regime and Thermal Energy

The initial substrate temperature was 28.4 °C, which was 6.5 °C higher than the ambient value, Figure 1a. A rapid increase in temperature to 45.9 °C was observed in the first 8 h (the heating rate was 2.2 °C h^−1^), then the temperature growth slowed four-fold, and over the next 10 h the temperature increased to 52 °C (0.6 °C h^−1^). The heating slowdown was caused by exceeding the temperature range of the tolerance of the microorganisms that dominate at this stage of the process and an insufficient number of thermophilic bacteria, which caused a further increase in temperature. The slowdown was followed by an increase to 64.6 °C on day 4 (the heating rate was 0.2 °C h^−1^) and a subsequent decrease to 60.3 °C on day 7. The temperature decrease was caused by a decrease in microbial activity due to the drying of the composted mass. Moistening on day 7 activated microorganisms that caused a temperature rise to a maximum of 69.0 ± 0.9 °C on day 8. A similar sharp temperature increase on days 14, 21, 28, and 56 also occurred due to moistening. On day 98, the substrate temperature approached the ambient temperature.

The heat of biological reactions was spent on heating the substrate and loss to the environment. The dynamics of cumulative thermal energy showed two zones when heat losses exceeded its formation, Appendix A. During the observations, 256.2 kJ of biological thermal energy was spent on increasing the temperature of 1 kg of substrate. The maximum heat release was observed in the first 4 h (4.28 kJ kg^−1^ h^−1^). During the first 87 h (3.5 days), the heat generation continuously exceeded the losses. In total, during this period, 60.81 kJ kg^−1^ was spent on raising the mass temperature (with an average of 0.7 kJ kg^−1^ h^−1^). There was an excess of biological heat generation over its losses to the environment for 571 h in the entire composting period, which was an average of 0.45 kJ kg^−1^ h^−1^.

#### 3.1.2. Gaseous Emissions

In the first 35 days of composting, the CO_2_ level in the exhaust air was 4.2 vol.% at an airflow rate of 0.04 L min^−1^ kg^−1^ dry matter (DM), Figure 1b. The microbial respiration rate (by CO_2_ formation) during the active period of composting (35 days) was 43.1 mg CO_2_ kg^−1^ h^−1^ (per DM) on average, and the maximum was 129.4 mg CO_2_ kg^−1^ h^−1^. On day 98, the microbial respiration rate was 4.6 mg CO_2_ kg^−1^ h^−1^.

In total, during the period of 98 days, the OM mineralization in CO_2_ form was 36,277 mg CO_2_ kg^−1^ DM (9795 mg C-CO_2_ kg^−1^ DM). The measured OM mineralization included the released CO_2_ from microbial decomposition and CH_4_ obtained in anaerobic processes. The value of CH_4_ in the exhaust air during composting was from 0.1 to 0.7 vol.%, with an average emission intensity of 1.4 mg CH_4_ kg^−1^ h^−1^. Overall, during the composting period, OM mineralization in CH_4_ form was 3251 mg CH_4_ kg^−1^ DM (2438 mg C-CH_4_ kg^−1^ DM). Thus, the mineralization of organic carbon in CO_2_ and CH_4_ form was in the ratio of 4:1 (9795:2438).

The NH_3_ emission was the highest on days 4–8 and ranged from 5.75 ± 0.46 to 5.87 ± 0.41 mg NH_3_ kg^−1^, with a maximum rate of 0.245 mg NH_3_ kg^−1^ h^−1^, Figure 1c. Rapid self-heating and reaching a temperature over 60 °C led to significant NH_3_ emissions. During the experiment period, nitrogen losses in NH_3_ form constituted 95.4 mg NH_3_ kg^−1^ DM. H_2_S emission was absent in the concentration that the device is able to detect.

Stirring (a–e in Figure 1a) provided the involvement of new substrate particles in the aerobic degradation process and consequently enhanced the microbial activity. Stirring and moistening of the substrate explain the wave pattern of temperature, CO_2_, and NH_3_ emission dynamics. Wong et al. [37] showed the same phenomenon during composting.

#### 3.1.3. Physicochemical Substrate Parameters

The OM content decreased by 59% from 64.81 ± 5.83 to 26.81 ± 1.88% in 98 days, Figure 2a. The most significant decrease (by 50%) occurred in the first 56 days. Total organic carbon (C) decreased in 98 days from 36.0 ± 3.2 to 14.9 ± 1.0%. Total Kjeldahl nitrogen decreased from 1.57 ± 0.20 to 1.15 ± 0.10% in 56 days and then increased to 1.68 ± 0.12%.

The maximum water loss, determined by weight loss, was observed in the first 28 days (209–352 g kg^−1^) due to the high-temperature conditions during this period. Wang et al. [38] also showed a significant decrease in water content (71 to 29%). The pH of the initial ms-OFMSW was low, at 5.5 ± 0.3, Figure 2b. Composting caused an increase in pH to close to neutral values (7–8). The EC of ms-OFMSW decreased in the first 14 days and then increased sharply, reaching the highest value on day 21, of 1839 ± 138 μS cm^−1^ (see Figure 2b). From day 56 to 98, the EC changed slightly from 989 ± 56 to 975 ± 87 μS cm^−1^. Low EC at the end of the experiment indicates the end of active OM mineralization.

The ammonium (NH_4_^+^) content significantly decreased from 653 ± 31 to 452 ± 46 mg kg^−1^ in the first 7 days of composting, Figure 2c. Later, an increase in the NH_4_^+^ content was observed on days 14–21, reaching a maximum value of 935 ± 62 mg kg^−1^. Subsequently, the ammonium content decreased to 38 ± 4 mg kg^−1^ on day 98. A similar trend was observed in the nitrate (NO_3_^−^) dynamics: during the first 7 days, NO_3_^−^ was not detected, and on day 21, its content reached a maximum value of 3812 ± 268 mg kg^−1^. The NO_3_^−^ content on day 98 of composting was 722 ± 64 mg kg^−1^.

#### 3.1.4. Stability Indicators

The C/N ratio of ms-OFMSW in the experiment decreased from 22.9 ± 1.4 to 8.9 ± 0.3, Figure 2d, due to significant OM mineralization and an increased nitrogen proportion to the maturation stage. This indicator is important for composting, both from a technological point of view and for the development of the microbial community. With a lack of nitrogen in the substrate, the rate of organic biodegradation decreases, and, conversely, an excess of nitrogen compounds leads to the emission of ammonia in a significant amount [39]. In addition, in our earlier work we were able to show that at high C/N, the biodiversity of the microbiota was lower than at lower values of C/N [17].

Initially, ms-OFMSW inhibited plant growth and development. As composting proceeded, the substrate phytotoxicity decreased, and it began to stimulate plant growth. Thus, GI increased from 69.5 ± 5.6% to 117.9 ± 4.7% in 98 days. The nitrification index (ratio of NH_4_^+^ and NO_3_^−^ content) reached a maximum value on day 56, of 13.87 ± 0.97. NI decreased to 0.05 on day 98, which indicated the BSRW (compost) stability at the end of the experiment [40].

### 3.2. Microbial Community Composition

#### 3.2.1. Fungal Community

Before composting (day 0), fungi from the phylum Mucoromycota (59%) dominated in ms-OFMSW. During composting, the phylum Ascomycota prevailed, with a relative abundance of at least 87%. A significant abundance of Basidiomycota was detected on days 0 and 7, of 5 and 10%, respectively. On day 0, 59% of the community was represented by the genus *Rhizopus*, and yeasts *Candida* (13%), *Clavispora* (3%), and *Kazachstania* (6%) were abundant (Figure 3). Thus, the yeast Saccharomycetes and the mold fungi Mucoraceae were the dominant groups in the initial ms-OFMSW.

The *Aspergillus* genus abundance was considerable on day 7 (32%) and, at the same time, fungi of the genus *Thermomyces* (23%) dominated. Based on the abundance of these organisms in the previous and subsequent periods, it can be concluded that *Aspergillus* belonged to the fungi of the heating stage and *Thermomyces* belonged to the high-temperature stage. The genera *Agaricus* (7%) and *Penicillium* (8%) can also be attributed to the typical fungi of the heating stage. On day 14, *Microascus* prevailed in the community (73%); the genera *Wardomyces* and *Pseudogymnoascus* comprised 5% of the fungal community each. All known *Pseudogymnoascus* strains grow at low temperatures. We could not exclude new, thermotolerant species discovery. Some fungi (e.g., *Wardomyces, Microascaceae, Geomyces*, and *Acrostalagmus*) were not detected at the onset of composting but evolved significantly during the process. These fungi may be introduced into the initial substrate in a non-detectable number or in the form of spores together with soil particles on the surface of potatoes, wood chips, and other components of MSW. Fungal spores are poorly detected by general molecular techniques. Clustering samples from different composting stages showed that the community composition on days 7–14 was significantly different from that on days 21–28. Similar communities were formed on days 21–56 when *Thermomyces* completely dominated (93–99%). At the beginning of the maturation phase (98 days), *Microascus* (24%) and *Thermomyces* (57%) prevailed in the ms-OFMSW fungal community.

#### 3.2.2. Prokaryotic Community

Phyla Firmicutes, Actinobacteria, Proteobacteria, Bacteroidetes, and Gemmatinomonadetes were detected in the prokaryotic community of ms-OFMSW. The abundance of *Firmicutes* gradually decreased with the composting time—from 89% on day 0 to 28% on day 98. The Actinobacteria abundance, on the contrary, increased during composting (up to 33–38% on days 21–98). By day 98, the proportions of Proteobacteria (15%), Gemmatinomonadetes (6%), and Bacteroidetes (6%) increased in the community. Archaea were poorly represented in the BSRW community. They accounted for less than 1% of the total abundance of prokaryotes. The reason for their low abundance was that archaea are usually oligotrophic and develop more slowly than bacteria [41] or develop under conditions of strict anaerobiosis (methanogens), which are created in a limited volume of compost. The latter is consistent with a low content of methane in the gas phase.

The initial ms-OFMSW community was significantly different from the remainder, which is clearly shown by the clustering of samples (Figure 4). The order Lactobacillales dominated in the feedstock microbiota: *Leuconostoc* (12%), *Weissella* (12%), *Lacticaseibacillus* (3%), *Companilactobacillus* (8%), *Lactiplantibacillus* (14%), *Levilactobacillus* (11%) and *Limosilactobacillus* (5%), Figure 4. Concurrently, *Klebsiella* (2%) and *Corynebacterium* (3%) were present in the substrate.

In the high-temperature stage, bacteria of the genus *Bacillus* actively developed. Their maximal abundance was on day 7 (33%), then the value gradually decreased over time. *Thermobifida*, *Streptomyces*, and *Planifilum* appeared in significant abundance on day 7 and were present in ms-OFMSW until the end of composting. The abundance of *Thermobifida* on days 7–28 varied within the range of 13–17%. The abundance of *Streptomyces* reached a maximum (16%) on day 21. The genus *Planifilum* comprised 16% on day 7, and then its abundance decreased to 5% on day 28. On days 28 and 56, *Oceanobacillus* (6–7%) and *Saccharomonospora* (5–6%) were considerably presented in the community. On day 56, *Bacillus* (12%), *Thermobifida* (14%), *Streptomyces* (7%), and *Planifilum* (6%) remained dominant. On day 98, the dominant genera in the community changed slightly. At the high-temperature stage, the cooling stage, and the beginning of maturation, the prokaryotic microbiota was similar.

#### 3.2.3. Potential Abilities of the Prokaryotic Community

At all composting stages, the main metabolic pathways involved in microbial growth and OM biodegradation were present in the microbiome (Appendix A). Despite the active aeration, methane metabolism genes were significantly presented in the community. According to KEGG analysis actinomycetes *Streptomyces* and *Thermobifida* made the main contribution to the metabolism of carbohydrates (starch and sucrose metabolism), methane, and fatty acid degradation at the high-temperature stage (Appendix A). Bacteria of *Bacillus* and SO134 (closely related to *Gemmatimonas*) also played a significant role in these metabolic pathways at the thermophilic stage. The detection of enzymes that degrade benzoate, the main intermediate in the degradation of aromatic hydrocarbons, and genes responsible for the conversion of aromatic hydrocarbons (styrene, toluene) indicates the potential activity of the studied communities throughout the experiment. For most communities, their potential ability to decompose styrene, toluene, naphthalene, chlorohydrocarbons, and other xenobiotics was revealed.

### 3.3. Microbial Biodiversity

The biodiversity of the microbial community was assessed by the Shannon and Chao1 indices and the OTU number, Table 2. The Shannon index of the prokaryotic community had the lowest value (2.52) on day 7, probably due to the death of certain prokaryotes when the temperature rose to 60 °C and higher. By day 98, the OTE number, and Chao1 and Shannon indices, significantly increased. The initial fungal diversity was high (Shannon index 3.15). Then, on days 14–28, all fungal biodiversity indicators decreased, probably due to an increase in the temperature, similar to the prokaryotic community. By day 28, the Shannon index decreased to 0.12. However, by day 98, the biodiversity of the fungal community recovered to the Shannon index value of 1.67.

### 3.4. Relationship between Physicochemical Processes and Microbial Dynamics

Pearson correlation coefficients (*p* < 0.05) were calculated to identify the relationship between the microbial community and the physicochemical parameters of composting (Appendix A). A positive correlation with the CO_2_ emission was observed for the genera *Bacillus*, *Aeribacillus*, and *Tepidimicrobium*. A positive correlation with C was shown for the genera *Novibacillus*, *Thermobifida*, *Saccharomonospora*, *Oceanobacillus*, and *Penicillium* and the Bacillaceae family. Temperature significantly correlated with the abundance of certain bacteria (*Paenibacillus*, *Aeribacillus*, *Ureibacillus*, *Vulgatibacter* and SO134). The abundance of the genera *Bacillus*, *Aeribacillus*, *Ureibacillus*, *Tepidimicrobium*, *Geobacillus*, *Planifilum*, *Penicillium*, and *Aspergillus* positively correlated with the NH_3_ emission. A positive correlation with pH was shown for the genera *Aeribacillus*, *Ureibacillus*, *Geobacillus*, *Oceanobacillus*, and SO134. The number of *Streptomyces*, *Paenibacillus*, and *Penicillium* positively correlated with the N content.

For the prokaryotic community, a significant negative relationship between biodiversity and the C/N ratio was shown. The prokaryotic Chao1 index negatively correlated with the OM and C content. Thus, while OM mineralization proceeded, a more complex and diverse community formed. The prokaryotic biodiversity increased, possibly due to a change in the range of organic substances, the reproduction of new organisms, and maintenance of the microbiota from the previous composting stages. Presumably, a C/N ratio decrease enhanced the development of various forms of prokaryotes due to an increase in the N content, which is a limiting factor for microbial growth. The Chao1 index of the prokaryotic microbiota positively correlated with the GI. The fungal biodiversity indices did not show a significant correlation with environmental parameters.

## 4. Discussion

The obtained results allow us to analyze the relationship between physical and chemical composting process parameters, and microbial community composition and biodiversity changes. The heating and activity of mesophilic microorganisms took place during the primary MSW processing during its accumulation, collection, transportation, and sorting. Thus, the high-temperature stage began much earlier than in similar research [38]. Furthermore, after about 35 days of active biodegradation, the cooling and maturation stages, final stabilization, and acquisition of BSRW followed.

### 4.1. Microbial Processes during the Substrate Preparation and Heating

Yeasts *Candida*, *Clavispora*, and *Kazachstania*, mold fungi of the genus *Rhizopus*, and lactic acid bacteria of the genera *Leuconostoc*, *Weissella*, *Lactiplantibacillus*, and *Levilactobacillus* were significantly represented in the initial ms-OFMSW. The genera *Penicillium*, *Aspergillus*, and *Agaricus* can also be attributed to the fungal dominants of the initial substrate. Though *Penicillium* is considered mesophilic, it has been detected during high-temperature composting by other researchers [42,43,44]. These microorganisms used easily degradable OM, whose content was initially high in ms-OFMSW. As a result of their metabolism, organic acids were produced that influenced the pH value on day 0. The predominance of acid-forming organisms in the community and a low pH at the beginning of composting were also shown in other studies [6,45,46,47,48]. In the research of Jurado et al. [7], Lactobacillales dominated the community throughout the entire industrial composting of OFMSW. In the current study, about a week passed from MSW collecting to the start of composting. During this period, due to the activity of facultative anaerobic microorganisms, an accumulation of acids occurred. Increased water content during the pretreatment of waste (it was not controlled in this stage) could also be a trigger for the development of yeast, *Rhizopus*, and lactic acid bacteria. In addition, during the experiment, with a total water content maintained at about 60%, some parts could have higher moisture due to the microzonality of the ms-OFMSW. In our early work, fresh food waste had a neutral pH value (6.9); then, in the first week of composting, the pH value decreased to 4.4 [17]. Therefore, the substrate acidification in the current work occurred before the composting started during the primary MSW processing due to the metabolism of the previously mentioned fungi and bacteria. The biodiversity indicators of the waste microbiota before composting were relatively high due to the favorable conditions for a vast range of microorganisms (aerobic, anaerobic, and microaerophilic conditions, moderate temperatures, and the maximal range of organic substances). As a result of the diverse microbiota activity, the temperature rose to 52 °C, and the CO_2_ level increased to 2 g CO_2_ day^−1^ kg^−1^ DM at the onset of composting.

### 4.2. Microbial Processes in the High-Temperature Stage

A sharp increase in temperature during the first day to 50 and 60 °C for 2–7 days triggered a significant change in the microbiota composition and contributed to thermophilic microorganisms, as also found by Ryckeboer et al. [14]. Under the influence of changes in temperature and the food source, microbial succession occurred. The prokaryotic microbiota diversity temporarily decreased on day 7, and on day 14, the fungal diversity significantly decreased. A rapid temperature rise and its maintenance for a long period (up to 28 days) were also revealed in research by Awasthi et al. and Zhang et al. [13,39]. The vigorous NH_3_ emission accompanied the decomposition of ms-OFMSW OM in the first week of composting. According to the correlations, the genera *Bacillus*, *Aeribacillus*, *Ureibacillus*, *Tepidimicrobium*, *Geobacillus*, *Planifilum*, *Penicillium*, and *Aspergillus* probably participated in ammonification. The fungal genera *Thermomyces*, *Agaricus*, *Aspergillus*, *Penicillium*, and *Microascus* that prevailed during the high-temperature composting stage formed various hydrolytic thermostable enzymes, and, thus, they actively participated in the decomposition of biopolymers [39,44,49]. During the high-temperature stage of the process, the typical prokaryotic groups were the genera *Bacillus*, *Thermobifida*, *Planifilum*, and *Streptomyces*. At the stage onset, a transition from the dominance of Lactobacillales to Bacillales occurred, as shown in the study by Graça et al. [6]. On day 7, *Bacillus* and related genera (*Ureibacillus*, *Aeribacillus*, and *Geobacillus*) mineralized the easily degradable OM and, according to correlations, increased the pH to slightly alkaline values. *Bacillus* species are known to secrete catabolic enzymes, such as proteases, which, through proteolysis, promote ammonification and contribute to pH increase [48]. Bacteria of the Bacillales order made a significant contribution to the decomposition of lignocellulose in the works by Antunes et al., Che et al., and Zhu et al. [45,50,51]. Thermophilic microorganisms *Thermobifida* and *Planifilum* were prominent participants in the recalcitrant OM decomposition and humus formation during composting [39,51,52]. During composting experiments by Xu et al. [53], *Thermobifida* was the main contributor to aerobic chemoheterotrophy, xylanolysis, cellulolysis, and methylotrophy. The Gemmatinomonadetes phylum was represented only by the SO134 group (closely related to *Gemmatimonas*), which was constantly present in the community, starting from the high-temperature stage. Lüneberg et al. [54] mentioned this microorganism in arid soils research. It is likely that its reproduction in the compost is associated with water loss caused by periodic evaporation. However, there is little information about *Gemmatimonas* and SO134 activity in compost; in the present work these bacteria contributed to the methane metabolism (probably to the methylotrophy process), and decomposition of carbohydrates and lipids. The *Streptomyces* genus was present during the high-temperature stage and cooling and was an essential participant in composting because its species can decompose lignin and cellulose [55,56] and secrete antimicrobial substances that inhibit the development of plant and human pathogens. The enzymes produced by *Streptomyces* have the potential to decompose toxic industrial and household waste [57].

As a result of the easily degradable OM oxidation and the decomposition of the polymers by the thermophilic bacteria and fungi, the most intense CO_2_ emission occurred at the high-temperature stage, which was also revealed by other researchers [58,59]. The mineralization of organic C and N positively correlated with the abundance of the genera *Bacillus*, *Aeribacillus*, and *Tepidimicrobium*. A positive correlation between OM content and abundance of microorganisms *Novibacillus*, *Thermobifida*, *Saccharomonospora*, *Oceanobacillus*, and *Penicillium* indicated their possible involvement in the decomposition of ms-OFMSW. The CH_4_ emission and a significant number of methane metabolism genes in the microbiome indicated the activity of an anaerobic community, which included the bacteria *Tepidimicrobium*, *Caldicoprobacter*, and *Limnochorda*. In the research by Che et al. [50], *Tepidimicrobium* and *Caldicoprobacter* contributed significantly to waste composting due to their ability to decompose proteins and carbohydrates.

During the first week, the decrease in the NH_4_^+^ content was associated with a high level of ammonia emission and an active process of assimilation of nitrogen compounds for microbial growth. On days 7–28, the NH_3_ emission decreased six-fold, apparently due to protein depletion as the primary source of released ammonia and activation of the nitrification process. The nitrification resulted in an increase in NO_3_^−^ content on days 7–21. The most significant nitrification activity during the high-temperature stage was revealed in our earlier research [17].

In the microbiome of ms-OFMSW, which contained many synthetic polymer compounds, there were a significant number of genes for the degradation of various xenobiotics. Therefore, ms-OFMSW may be suitable for isolating relevant microorganisms for study and further biotechnological application.

### 4.3. Microbial Processes at the Cooling and Maturation Stage

*Microascus* and *Thermomyces* continued to dominate during cooling and maturation, which indicated the ongoing decomposition of biopolymers, especially plant residues. At cooling and maturation stages, the composition of the prokaryotic community also differed little from that during the high-temperature stage. *Bacillus*, *Planifilum*, *Streptomyces*, *Thermobifida*, etc., had significant abundance during composting due to these microorganisms’ broad functional activity range, aimed at the decomposition of easily degradable and recalcitrant OM. The decrease in the NH_4_^+^ level occurred due to the end of active ammonification because most of the proteins and amino acids were consumed. Possibly, nitrogen fixation contributed to an N content increase at the late composting stages. According to the correlation analysis, *Streptomyces* and *Paenibacillus* may be involved in the nitrogen fixation process. Nitrogen-fixing microorganisms have been detected within the genus *Planifilum* and *Thermobifida*, and the *Bacillaceae* family (including the genus *Paenibacillus*), in composting research [55,60]. Thus, nitrogen accumulation may occur due to a significant range of bacteria that make up the compost community. The accumulation of nitrogen enhanced the biodiversity of the prokaryotic community. Due to the use of various energy substrates of compost and, consequently, the growth and reproduction of microorganisms, the prokaryotic community at the maturation stage was the most diverse, which was revealed in the works by Galitskaya et al., Graça et al., Mironov et al., and Ryckeboer et al. [6,14,17,47]. As a result of the activity of fungi and prokaryotes during aerobic solid-phase waste processing, a stable, non-toxic compost, which stimulates plant growth, was obtained.

## 5. Conclusions

For the first time, the research of a successive change in the microbiota composition during ms-OFMSW composting was conducted.

Ms-OFMSW contains a fully functional microbial community, which gradually evolves under the influence of physical and chemical factors and effectively decomposes the organic matter of the waste.

The initial biodegradation processes occurred due to the metabolism of lactic acid bacteria *Leuconostoc*, *Weissella*, *Lactiplantibacillus*, and *Levilactobacillus*, mold fungi *Rhizopus*, *Agaricus*, *Penicillium*, and *Aspergillus*, and yeasts *Candida*, *Clavispora*, and *Kazachstania*.

The main biodestructors at the high-temperature stage were bacteria of the genera *Bacillus*, *Streptomyces*, *Thermobifida*, and *Planifilum*, and fungi of the genera *Thermomyces* and *Microascus*. The same microorganisms were predominant in the ms-OFMSW during cooling and maturation, probably due to the wide range of degradation capabilities.

Anaerobic degradation zones were formed in ms-OFMSW when oxygen was significantly consumed, confirmed by methane emission, methane metabolism genes in the microbiome, and anaerobic hydrolytic bacteria (*Tepidimicrobium* and *Caldicoprobacter*) in the community.

The potential high functional diversity of the prokaryotic community was revealed. Therefore, ms-OFMSW can be used to isolate microorganisms that are biodestructors of xenobiotics (plastic, among others), and to create starter cultures to intensify the ms-OFMSW composting process.

## Figures and Tables

**Figure 1 microorganisms-09-01877-f001:**
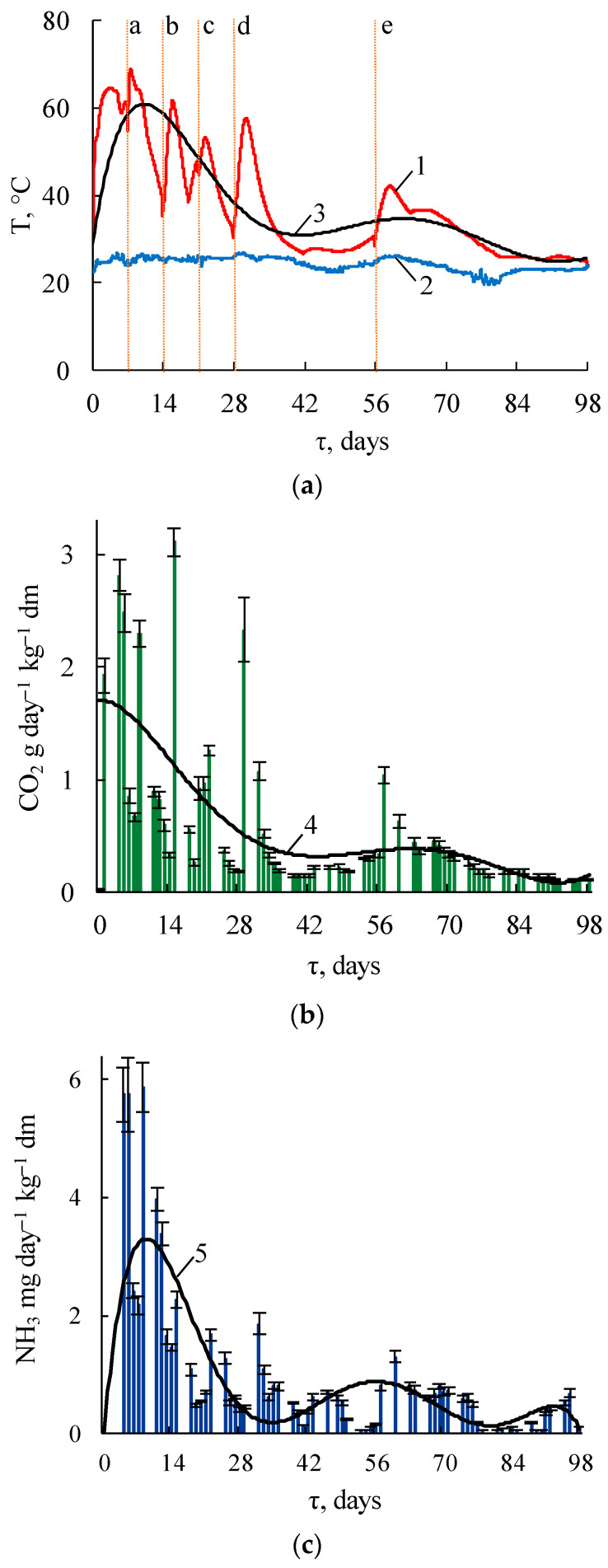
The temperature regime and gaseous emissions. (**a**) temperature: 1—substrate; 2—ambient; 3—approximation of the substrate temperature (R^2^ = 0.67 *p* < 0.05); a–e—stirring and moistening; (**b**) CO_2_ emission: 4—approximation (R^2^ = 0.4 *p* < 0.05); (**c**) NH_3_ emission: 5—approximation (R^2^ = 0.54 *p* < 0.05).

**Figure 2 microorganisms-09-01877-f002:**
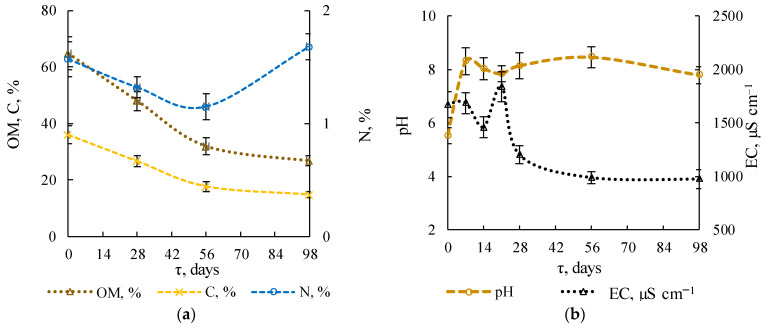
Changes in chemical and physical parameters and stability indicators: (**a**) relative OM, C, N content; (**b**) pH and EC; (**c**) NH_4_^+^ and NO_3_^−^ content; (**d**) stability indicators: GI, NI, C/N.

**Figure 3 microorganisms-09-01877-f003:**
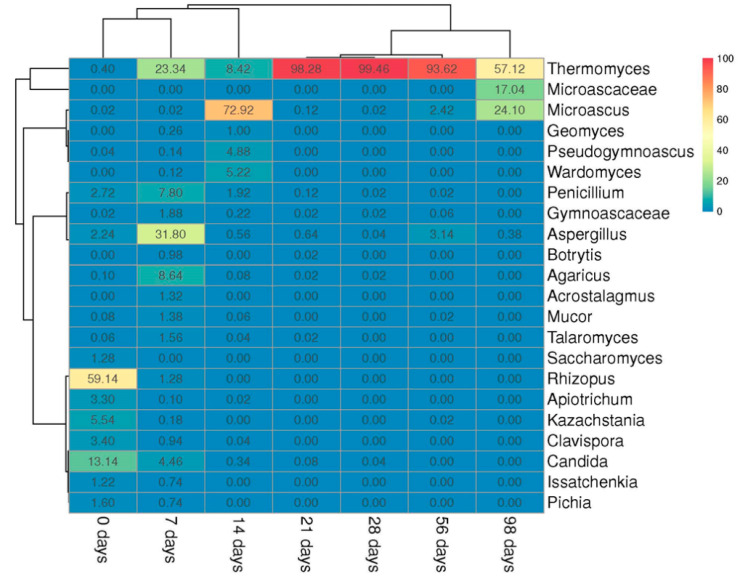
The heat map of the distribution of the most relatively abundant (>1%) fungal genera on days 0, 7, 14, 21, 28, 56, and 98 of composting. The genus abundance was calculated by dividing the sequence proportions by the total sequence count in the sample. The relative abundance of the genus is shown by the color.

**Figure 4 microorganisms-09-01877-f004:**
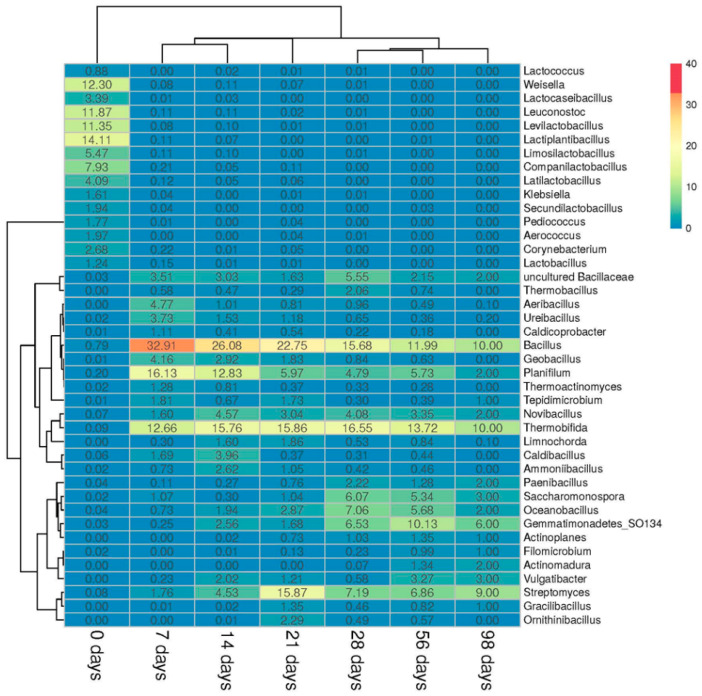
The heat map of the distribution of the most relatively abundant (>1%) prokaryotic genera on days 0, 7, 14, 21, 28, 56, and 98 of composting. The genus abundance was calculated by dividing the sequence proportions by the total sequence count in the sample. The relative abundance of the genus is shown by color.

**Table 1 microorganisms-09-01877-t001:** The principal physical, chemical, biological, and physicochemical parameters of the initial ms-OFMSW.

Parameter	Units	Value *	Optimal Limits
pH	pH units	5.5 ± 0.3	6.5–8.0 [22]
Electrical conductivity (EC)	µS cm^−1^	1670 ± 124	
Water content	%	60.26 ± 3.01	50–60% [22] 40–50% [23]
Total Kjeldahl nitrogen (N)	%	1.57 ± 0.16	
NH_4_^+^	mg kg^−1^	653 ± 31	
NO_3_^−^	mg kg^−1^	0.0 ± 0.0	
Organic matter (OM)	%	64.81 ± 5.83	
Total organic carbon (C)	%	36.0 ± 3.2	
C/N		22.9 ± 1.4	25–30 [11,22,23]
Germination index (GI)	%	69.5 ± 5.6	

*—average value ± standard deviation of three replicates.

**Table 2 microorganisms-09-01877-t002:** Biodiversity of prokaryotic and fungal communities.

Sample, Days	Number of OTUs	Chao1	Shannon Index (Path)
Prokaryotes			
0	1166	3944	3.07
7	1324	4289	2.52
14	1193	5143	2.70
21	1461	5221	2.85
28	1146	3486	3.05
56	1368	5700	3.38
98	1781	7724	4.13
Fungi			
0	215	266	3.15
7	351	398	4.95
14	145	218	1.88
21	96	176	0.28
28	39	95	0.12
56	75	125	0.63
98	82	151	1.67

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
