# Peer review of "Microbiota Dynamics of Mechanically Separated Organic Fraction of Municipal Solid Waste during Composting"

_microorganisms, 2021, doi:10.3390/microorganisms9091877_

Round 1
Reviewer 1 Report
Dear reader,
The focus of review will be from my point the microbial part. As far as I am able to check the chemical analysis, this shows high potential.
However, the microbial part is in my opinion short and needs elaborate experiments to show actual microbes to be present and growing.
In that perspective, cultivation experiments should be performed besides the sequencing to see that certain species are present during compostation. To elaborate on this, the following comments are made;
- As indicated in the materials and methods, the experiment was performed in a lab. The observation of bacteria and fungi appearing on the substrates in a lab environment does indicate that the bacteria and/or fungi are not originating from the material itself (this then should appear at t=0) but from other environmental causes. This is the case for the Geomyces, Acrostalagmus, Wardomyces and Microascaceae, which are absent at t=0 and appear at a later time point. The presence of these fungi should at least be discussed in the discussion
- The presence of thermophilic bacteria and fungi is expected at elevated temperatures. However, the presence of Penicillium and Pseudogymnoascus species is remarkable as these grow mainly at cold climates with temperatures ranging from 0 to a maximum of around 30C for most species.
- The data show that the DNA of these fungi are present, but they are likely not to be viable (which could be proven by cultivation).
- The presence of other types of fungi indicate that the humidity of the sample at sometimes was high; aw >85% (indicated by presence of Botrytis and Rhizopus strains), which conflicts with measurements performed.
The material and methods describe the search for genes of degradation of bacteria and fungi by using KEGG, however, this is barely described in the results and/or supplementary data.
The material and methods describe that the experiment was performed in triplicate, was this also done for the sequencing and OTU assignments?
Was the amount of generated sequences of ITS (10000 as indicated) only sufficient to detect 39 OTUs as t= 28 days? This sounds as a really low number and high abundance of only one fungus.
In conclusion, the article is unable to answer both objectives; the composition and function of microbial communities is based on the data not complete. The relationship of microbial diversity compared to environmental parameters is not sufficiently described.
Reviewer 2 Report
Firstly, the manuscript describes the study of mechanically separated organic fraction of municipal solid waste (ms-OFMSW) from physico-chemical perspectives to microbial diversity perspectives. It is quite interesting that the microbiota of the fungal community and prokaryotic community varied and have different relationships with the physico-chemical properties. Discussion on the microbial communities along the processes and addressing the key players by the processes in the Conclusion section helped for better understanding the results.
Secondly, it would be good to address the answers for some questions:
(1) the higher is biodiversity, the better is for composting process and recover/degrade specific materials?
(2) otherwise, would it be advantageous to have a few dominant species in the diversity?
(3) how can you develop the microbial community for the specific target; for instance, what will be the important factors for optimization to biodegradation of xenobiotic materials?
Finally, if the following comments are reflected, the manuscript could be more improved.
Line #210 and 228-229.
A sentence or two does not seem to be good as a paragraph.
Line #220.
Figure 1a should be adjusted its visual quality to be similar to others.
Line #336 and 340.
Typo: S0134 -> SO134
Line #463.
It would be better to put the word “ms-OFMSW” than to put “Mechanically sorted OFMSW”.
Table S2 in the Supplementary Materials.
Typo: uncultered -> uncultured
Subscript: CO2 -> CO2
Reviewer 3 Report
The study of the succession of the total microbiota (bacteria, archaea, fungi) and the emission of gases during the composting of ms-OFMSW is of absolute interest. The paper is carried out at a high scientific level and deserves publication in the Journal.
Notes and Questions:
- For the experiments a material volume of 10 cubic decimeters was used. Is this enough to correctly represent the heterogeneous substrate of municipal solid waste? It would be possible to more fully characterize the initial substrate by evaluating the amount of carbohydrates, proteins, lipids and other organic substances in it. Thus, the merit of the paper, emphasized by the authors, is that working with real MSW material, is at the same time its disadvantage.
- Line 101: How was 60% moisture measured?
- Table 1. It is necessary to provide methods for measuring pH and C / N ratio, not just a reference to previous work. Was aqueous or saline pH measured?
- Lines 115-126: Was the external temperature taken into account when making the calculations, or was the experiment carried out under thermostatically controlled (or controlled) conditions?
- Section 2.5. How and in what quantity were samples of the substrate taken for the analysis of the microbial community?
- Lines 179, 314, 332: Why are the numbers of figures and tables indicated as S1 and S2?
- Figure 1. Gas emissions are shown with standard deviations and temperature changes without fluctuation intervals.
- Line 242: The C / N ratio is more commonly used in the agrochemical literature. Therefore, the correlation with this indicator is not entirely informative. It is also necessary to provide in the appropriate section a description of the method for determining this ratio.
- Lines 245, 246: Nothing is said about growing plants on a substrate in the Materials and Methods section.
- Section 3.3. Moistening was carried out before the sampling of the substrate for the analysis of diversity?
